# Process Parameter Optimization for Hybrid Manufacturing of PLA Components with Improved Surface Quality

**DOI:** 10.3390/polym15173610

**Published:** 2023-08-31

**Authors:** Sergiu Pascu, Nicolae Balc

**Affiliations:** Department of Manufacturing Engineering, Faculty of Industrial Engineering, Robotics and Production Management, Technical University of Cluj-Napoca, Memorandumului 28, 400114 Cluj-Napoca, Romania; sergiu_pascu2002@yahoo.com

**Keywords:** hybrid manufacturing, biodegradable thermoplastic polymer, PLA components, process parameters optimization, roughness prediction, neural network modeling

## Abstract

This paper presents a new method of process parameter optimization, adequate for 3D printing of PLA (Polylactic Acid) components. The authors developed a new piece of Hybrid Manufacturing Equipment (HME), suitable for producing complex parts made from a biodegradable thermoplastic polymer, to promote environmental sustainability. Our new HME equipment produces PLA parts by both additive and subtractive techniques, with the aim of obtaining accurate PLA components with good surface quality. A design of experiments has been applied for optimization purposes. The following manufacturing parameters were analyzed: rotation of the spindle, cutting depth, feed rate, layer thickness, nozzle speed, and surface roughness. Linear regression models and neural network models were developed to improve and predict the surface roughness of the manufactured parts. A new test part was designed and manufactured from PLA to validate the new mathematical models, which can now be applied for producing complex parts made from polymer materials. The neural network modeling (NNM) allowed us to obtain much better precision in predicting the final surface roughness ($R_{a}$), as compared to the conventional linear regression models (LNM). Based on these modelling methods, the authors developed a practical methodology to optimize the process parameters in order to improve the surface quality of the 3D-printed components and to predict the actual roughness values. The main advantages of the results proposed for hybrid manufacturing using polymer materials like PLA are the optimized process parameters for both 3D printing and milling. A case study has been undertaken by the authors, who designed a specific test part for their new hybrid manufacturing equipment (HME), in order to test the new methodology of optimizing the process parameters, to validate the capability of the new HME. At the same time, this new methodology could be replicated by other researchers and is useful as a guideline on how to optimize the process parameters for newly developed equipment. The innovative approach holds potential for widespread equipment functionality enhancement among other users.

## 1. Introduction

Additive manufacturing operations such as fuse deposition modeling (FDM) are frequently used for modeling, small application batches, and realizing small-to-medium prototypes. Since this is a manufacturing operation with a smaller cost compared to the other options, this is used more frequently for obtaining 3D-printed parts. It is well known that the accuracy and the surface finish of 3D-printed parts are not god, but these results, to some extent, are correlated with the printing parameters. In cases in which the obtained result is not the desired one, although the input parameters were chosen based on similar studies or on experience, additional operations need to be performed. Operations like milling a 3D-printed part can highly improve the roughness and the accuracy of the final part. Thereby, the hybrid manufacturing concept is introduced. Both technologies have their own limitations. The milling process has limitations related to complexity, making milling not suitable for the size of the cutting tool, which can result in additional processes to complete the part. Milling is not suitable for all types of materials, especially those that are soft or brittle, as they can easily deform or break under the pressure of the cutting tool. It is an expensive process, as it requires specialized equipment and skilled operators.

On the other hand, FDM printing has limitations on the resolution of the parts it can produce. The quality of the final product can be limited by the layer height and the accuracy of the printer.

Herein, samples were manufactured from PLA (Polylactic Acid), which is widely used to produce complex components made from polymer materials. PLA is a popular choice in additive manufacturing due to its biodegradability, ease of processing, and reasonable costs. PLA has its own limitations in terms of surface finish, dimensional accuracy, and overall quality of the complex parts made by 3D printing. Through the proposed hybrid manufacturing approach—the combination of fused deposition modeling (FDM) printing with a milling operation—the researchers aim to address these limitations and improve the surface quality and dimensional accuracy of PLA-based parts.

The primary motivation driving this research is the exploration of the extent of effectiveness of hybrid manufacturing (3D printing combined with milling) for producing complex parts made from PLA materials.

The objectives of the work are centered on testing the feasibility of using PLA in a hybrid manufacturing process and on validating the new equipment and methodology by optimizing the process parameters. Specific objectives are to develop predictive models for surface roughness and to test and validate the new methodology in practical case studies, which might be reproduced by other researchers working in similar environments, aiming to enhance the functionality of their newly developed equipment. The main objective is to determine and test, experimentally, whether polymer materials, specifically, PLA, are suitable for a hybrid manufacturing process which combines 3D printing and milling operations. This involves assessing the surface roughness and accuracy of the produced PLA components. Another objective is to validate the capability of the new Hybrid Manufacturing Equipment (HME) through a case study involving a complex test part realized from PLA. This aimed to demonstrate the efficiency of the equipment and to provide potential for replicating this methodology in similar setups.

## 2. State of the Art

Hybrid manufacturing (HM) technology has witnessed remarkable advancements in recent years but, at the same time, came with associated drawbacks. The aspects related to the drawbacks of both techniques can be minimized by combining both technologies [1,2]. A.N.M. Amanullah, in [3], designed and manufactured a similar type of equipment that was able to perform printing and milling operations on the same structure. The innovation was installing the computer numerical control (CNC) cutting spindle and the heated extruder of the FDM on a rotary structure and using infrared sensors (IR) which simplified the mechanism structure and overcame the problem of misalignment of the manufacturing tools.

The incorporation of five axes in this machine brings several benefits, which were showcased [4]. For instance, the five-axis machine successfully created overhang features without the need for support material. This not only reduced costs and build-time associated with support, but also minimized the risk of deformation caused by support material removal. Moreover, the five-axis hybrid system enabled the production of FDM parts with embedded material, a task that cannot be achieved on a standard three-axis FDM machine. The inclusion of embedded material effectively enhanced the stiffness of the FDM item while reducing manufacturing time and costs. Furthermore, strategic use of the embedded material could improve stability by lowering the center of gravity of the FDM part. The hybrid system also demonstrated its capability to trim freeform surfaces and enlarge tapered holes in FDM parts, which cannot be accomplished using traditional three-axis RP machines. Notably, the hybrid technique significantly reduced build-time by 44% and 58% in two specific cases. The advantages offered by the proposed hybrid system, combining FDM and five-axis machining, were evident and impactful.

In terms of surface quality, the *R_a_* parameter, used as an indicator in this study [5], shows superior performance at lower feed rates (400 mm/min) and a low depth cut (0.2 mm) across all studied rotational speeds. On the other hand, when considering burr height, the minimum heights are achieved by employing a higher feed rate (800 mm/min), a higher level of depth cut (0.8 mm), and a relatively high rotational speed. This phenomenon can be attributed to the heat exchange during the milling process. As the tool remains in contact with the sample surface for a longer duration, it leads to an increase in temperature and subsequently an increase in burr height. Furthermore, the optimal milling parameters for achieving better surface quality differ from those that result in lower burr formation. It was discovered that, when pooling was not used, only the thickness of each layer had a significant effect of 49.37% at a 95% confidence level. However, when pooling was employed, the layer thickness had an increased effectiveness of 51.57% at a 99% confidence level. Additionally, the factors of material width and printing speed contributed significantly, accounting for 15.57% and 15.83% respectively, at a 99% confidence level. The correlation analysis further emphasized the importance of layer thickness, demonstrating a strong inverse relationship with surface roughness.

Based on the analysis, it was determined that the layer thickness had the greatest impact when set at level 3 (0.3556 mm), while the road width was most effective at level 1 (0.537 mm), and the deposition speed at level 3 (200 mm). Bintara, in [6], studied and examined the surface roughness and production duration as key parameters. The analysis revealed that the layer height plays a significant role in determining surface roughness. The smallest roughness was observed at a layer height of 0.05 mm, while the lowest roughness occurred at a layer height of 0.25 mm. Furthermore, optimal printing parameters were identified at layer heights of 0.15 mm and 0.2 mm.

Based on the research conducted by Pamarac in [7], the optimal parameters were identified for milling 3D-printed components while maintaining a consistent spindle speed of 3500 revolutions per minute. In the case of PLA, a higher cutting speed leads to improved surface quality. This distinction arises from the fact that PLA has a lower melting temperature. Consequently, the longer the cutting tool remains in contact with the component, the more heat is generated, resulting in a degradation of the part’s surface. In another article [8], the researchers aimed to study the behavior and the roughness of the surface by also including the printing angle. It was found that a measuring direction of 90° yielded the most representative value for the distribution of surface roughness (Ra), compared to measuring angles of 0° and 45°. Decreasing the nozzle diameter from 0.3 mm to 0.2 mm and reducing the layer height to 0.1 mm resulted in a significant increase in both the build-time and the consumption of thermoplastic filament for printed parts number 5 and number 6. In terms of the inner faces of the printed parts, the highest surface roughness behavior was observed at a 45°-measuring direction, with a value of 36.96 ± 0.51 µm, while the lowest surface roughness behavior was recorded at a 0°-measuring direction, with a value of 1.35 ± 0.27 µm. These measurements were obtained using a nozzle diameter of 0.5 mm and a layer height of 0.3 mm.

For the outer faces, the highest surface roughness behavior was measured at a 45°-measuring direction, with a value of 34.94 ± 1.04 µm, using a nozzle diameter of 0.5 mm and a layer height of 0.3 mm. Conversely, the lowest surface roughness behavior was found at a 0°-measuring direction, with a value of 1.08 ± 0.30 µm, using a nozzle diameter of 0.3 mm and a layer height of 0.2 mm. Comparing the inner and outer faces of all six configurations of FDM 3D-printed parts, it was observed that the surface roughness behavior differed by approximately 2 µm (5%) for higher values. Conversely, for lower values of surface roughness behavior, the difference between the inner and outer faces was approximately 0.27 µm (20%).

In another study [9], the impact of technological parameters on surface roughness, material deposition, and hybrid manufacturing time was investigated using a statistical approach. Regression models were analyzed to determine the optimal values for parameters such as spindle speed, layer height, material compensation flow, printing speed, feed speed of the milling tool, and milling depth. The findings showed that layer height had the most significant influence on surface roughness, with higher layers resulting in minimal roughness. The material compensation flow had no effect on roughness at the highest layer height. A high spindle speed could cause material winding on the tool’s surface, so a lower value was deemed optimal. The feed speed of the milling tool had an inverse relationship with roughness. The impact of milling depth varied depending on the nozzle size. The material deposition was primarily influenced by the material compensation flow, while hybrid manufacturing time was minimized by using the fastest printing speed and highest layer height. In conclusion, using a larger nozzle size allowed for achieving the same final surface roughness as the smaller nozzle size but with significantly higher productivity. His research aimed to investigate the surface roughness of 3D-printed parts manufactured using a low-cost desktop FDM 3D printer and PLA+ thermoplastic filament.

In another study [10], the researchers used neural model optimization to predict surface roughness and it was observed that the artificial neural network models exhibited greater accuracy in predicting surface roughness. By analyzing the Root Mean Square Error and Mean Absolute Percentage Error values, it was evident that the neural model Multilayer Perceptron (MLP) 4-15-1, incorporating four input variables (three cutting parameters and machining vibration), displayed a higher average prediction accuracy of 93.14% compared to the model MLP 3-13-1, which only considered three cutting parameters. These findings provide clear evidence that incorporating spindle vibration into the prediction algorithm can enhance the accuracy of predictions. D. Kramar, in [9], used regression models to identify the optimal values for various parameters, including spindle speed (n), layer height (h), material compensation flow (Φ), printing speed (v), feed speed of the milling tool (vf), and milling depth (ap). Regarding surface roughness, the primary factor influencing roughness was found to be the layer height (h). Achieving the highest layer height resulted in minimal roughness ($R_{a}$ and $R_{y}$) when measuring in different directions. At this specific layer height, the material compensation flow (Φ) was observed to have no significant impact on roughness. Additionally, a higher spindle speed (n) was found to lead to material winding on the tool’s surface, so a lower value was considered optimal. The feed speed of the milling tool (vf) exhibited an inverse relationship with roughness. The depth of milling (ap) only had a marginal influence, with a smaller milling depth contributing to better roughness when using a smaller nozzle size while having a minimal impact with a larger nozzle size. In terms of material deposition during hybrid manufacturing, the primary factor influencing deposition (MD) was found to be the material compensation flow (Φ), which had a proportional effect.

## 3. Mathematical Modeling by Linear Regression of Printing and Milling

For a better understanding of the process, Figure 1 and Figure 2 present a workflow that represents the steps performed during this research to find and optimize the manufacturing parameters for milling and printing operation.

The authors analyzed previous research [9] that dealt with a similar task of looking to improve the process parameters in order to obtain a better surface quality of the PLA parts. This is where our first idea came from, as to how to select the important parameters and how to collect the data. We kept constant the nozzle size and took into account the specific conditions of our new HME. Other reasons leading to select these variable parameters to be optimized were related to other preliminary experiments conducted using this new HME, which showed which of the functional parameters had a significant influence on the result of the fabrication. Such preliminary experiments made by the authors have been published before, in [11,12].

Data collection for the milling operation included preliminary experiments using the following input parameters:Spindle rotational speed n [rotation/min];Cutting depth ap [mm];Feed rate vf [mm/min].

Data collection for the printing operation included preliminary experiments having the following input parameters:Layer thickness h [mm];Nozzle speed *v* [mm/s];Flow rate compensation Φ [%].

While printing the samples, a standard nozzle of 0.5 mm was used and, for the milling tool, a high speed (HSS) milling tool of 6 mm diameter with 4 teeth was used.

The focus of the study was to have controllable factors; in our case, the milling and the printing parameters were used to establish, optimize, and predict the uncontrollable factors (also called noise factors) like roughness. The technological parameters were examined on three levels: −1 is the lowest value of the parameter, +1 is the highest value of the examination parameter, and 0 represents the average between the low −1 and the high +1. The values of technological parameters in 3 levels are presented in Table 1.

The analysis for optimizing the manufacturing parameters was performed separately for the printing operation and for the milling operation, in order for each operation to be studied individually.

The Taguchi L16 method was used to efficiently identify the optimal combination of parameter settings in hybrid manufacturing by conducting a limited number of experiments, analyzing factor interactions, and improving product quality by minimizing surface roughness. The decreased number of preliminary experiments, both for printing and milling, in order to select the relevant necessary experiment are presented in Table 2 and Table 3.

The output parameter Ra [µm] represents the roughness values required for model validation and represents the measurement value for each experimental set. The roughness measurements were performed with a Mitutoyo Contracer CP—200/400—Contour Measuring Instrument. The measurement parameters used were as follows: the measurement range was set to +/−500 µm, in accordance with the ISO 25178-7:2010 standard [13]; the measurement length was set at 12mm; and the measurement speed was maintained at 0.3 mm/s. The cutoff value used was Gaussian.

Figure 3 and Figure 4 illustrate the hybrid prototype engaged in the printing and milling operation. The new HME prototype designed by the authors requires specific sets of parameters both for milling and printing, which depend on specific components of the HME, which had a special calibration procedure. That is why process parameter optimization is required to provide the best surface quality and to reduce material consumption.

### 3.1. Data Analysis and Model Generation

After performing all 32 experiments on the hybrid and all the necessary output data were collected, the values were uploaded into Project Modeling 5.3 Software for further analysis. After the data were imported, the first step was to trend the whole batch of data. This action was required in order to see if there were any anomalies that could affect the model. This step of cleaning the data is necessary before running the model algorithm. Once the cleaning operation was performed, the data were imported into a Linear Regression Model (LRM) to be analyzed. The goal of linear regression is to find the best linear relationship between the variables that minimizes the sum of squared errors. The results of the model are illustrated in Figure 5 for the printing operation and in Figure 6 for the milling operation. For a more visual understanding of the results, a trend representation was made with the actual roughness of the sample and the estimated roughness of the model for both cases.

As can be observed from both trends, there is a slight variation in the trends when comparing the measurement roughness and the estimated one. The mathematical formulas of both models were generated by the software application based on the input and output parameters. The $R_{a}$ estimate was generated by applying the formula to the input parameters. Equation (1) represents the mathematical model used to determine the roughness obtained for the printing operation, while Equation (2) represents the mathematical model used in the determination of the roughness for milling. These equations can estimate the final roughness of the part when requirements or limitations like the rotation of the spindle are involved.
*R_a_* [µm] = 0.00606331815233892 ∗ v [mm/s] + 0.801571903203949 ∗ h [mm] v 0.0107905747401297 ∗ Φ [%] + 1.36(1)
*R_a_* [µm] = 3.87368421052632 × 10^−5^ n [rot/min] − 0.00199899384469697 ∗ vf [mm/min] − 0.26022727272727 ∗ ap [mm] + 0.8(2)

By analyzing the accuracy of the model, the value of the average relative error was 0.13% for printing and 0.38% for milling. Theoretically, this is considered too high since the value needs to be as close as possible to 0. Another important indicator was the F-test, which measures if the overall regression model is statically significant. A high F-statistic and a low *p*-value indicate that the model is a good fit for the data and that the independent variables included in the model are significant predictors of the dependent variable. In our case, the F-statistic and *p*-value for printing were 1.20% and 0.349%. The same indicators for milling were 0.48% and 0.85%. The R-squared indicator, whose value needs to be as close as possible to 1, had, in this case, values around 0.13–0.38%. These indicators show that the model may be considered moderately accurate, but not highly accurate. The average relative errors of 0.38% and 0.13% indicate that, on average, the model’s predictions are off by about 31% and 13%, respectively, from the actual values. This means that the model’s predictions may have a significant margin of error, which could limit its usefulness.

#### Model Validation

To perform a validation of the mathematical model for both operations, workflow diagrams were established, shown in Figure 7 for the operation process and Figure 8 for the milling operation.

In practice, the user knows the required surface quality of the part which has to be obtained and needs to set up the optimal parameters of the HME in order to obtain the part with sufficient surface quality. Starting from the input and output values presented in Table 2 and Table 3, we generated, in Project Modeling Studio Software, five sets of optimized parameters for each operation. The parameters are presented in Table 4. These actions were required in order to validate our mathematical model. The next step of the validation process was to manufacture the samples according to Table 4, where Te. $R_{a}$ represents the theoretical roughness and Ef. $R_{a}$ represents the effective roughness.

After manufacturing, a new set of roughness measurements were performed, and new data were obtained. In Table 4, Te. $R_{a}$ is the theoretical value obtained using Equations (1) and (2). The final step was to perform the validation of the linear regression model using the optimized parameters. For this validation step, the values of the effective roughness were compared with the value of the theoretical roughness. By analyzing the roughness value, it can be seen that the value of the effective roughness decreased as compared to the previous values before optimization. The optimization was based on choosing to modify the correlated inputs which were influencing the manufacturing operations. Because none of the input parameters were highly correlated with the roughness, all the parameters were taken in the analysis for improvements in the model performance. To provide a better understanding of the small differences between theoretical roughness Te. $R_{a}$ and the effective roughness Ef. $R_{a}$, a trend using their values is provided in Figure 9. It can be seen that the average relative error of the model is 18.1%. Figure 10 presents the validation trend for milling operations and also, in this case, the measured roughness was greater than the theoretical values.

The mathematical model has some small deviations from the effectively measured roughness. After conducting a comparative analysis of the theoretical roughness and effective roughness, it was determined that the mean absolute error (MAE) between the predicted values and the actual values was 0.16/0.13 µm. This result may require further refinement to improve the accuracy.

## 4. Neural Network Modeling to Improve the Linear Regression Model

Starting from the assumption that the linear regression model of this study can be further optimized, the same batch of samples was used for building the model. The neural network model is a more complex and flexible method that can capture non-linear relationships between variables. It involves a network of interconnected nodes that process and transmit information. The Levenberg–Marquardt algorithm was used as an optimization algorithm. This algorithm is one of the methods used for training a neural network model, which involves adjusting the weights between the nodes to minimize the error between the predicted and actual values. By maximizing the power of neural networks and by using the Levenberg–Marquardt algorithm, complex patterns can be modeled and more accurate predictions cab be obtained from the available data. This approach is particularly valuable when dealing with small datasets where linear regression may not adequately capture the underlying relationships between variables. Due to the small number of tests performed, the approach used was 80&20, which means that the whole data set was split in two groups. One had 80% of the data, which was used to train the model. During this training phase, the model learned from the data’s patterns, relationships, and characteristics. The remaining data (20%) were used to validate the model’s performance. This split ensures that the validation data are completely independent of the training data, simulating real-world scenarios where the model needs to make predictions on new, unseen data. This is a well-known and widely used approach in the data analytics area.

### Neutral Network Model Generation

The input and output data were again imported into the Project Modeling Studio Software. Some of the most important parameters for this method to generate an accurate model are the number of iterations, learning rate, and hidden layers in the network. The number of iterations refers to the number of times the optimization algorithm updates the weights and biases of the network in order to minimize the loss function. The optimization process involved an iterative approach to adjust the weights and biases of the neural network to minimize the loss function. The loss function measures the difference between the predicted output of the neural network and the true output. The goal is to minimize this difference by iteratively adjusting the weights and biases of the network. The total number of iterations considered was 5000. The learning rate is the training parameter that controls the size of weight and bias changes during the learning process. The value of this parameter was selected as 0.1. The hidden layers are layers of artificial neurons that are located between the input layer and the output layer. The term hidden refers to the fact that the operations performed by the neurons in these layers are not directly visible or accessible from outside the network. The input layer of a neural network receives the input data, which is then processed by the hidden layers before being passed to the output layer. Each neuron in a hidden layer receives input from the neurons in the previous layer, applies a mathematical function to the inputs, and produces an output that is passed on to the next layer. The value of this parameter was set to 2. For both neutral network models that were analyzed, the settings were the same—only the input parameters were different. Once the neutral network models were generated, the results were analyzed, as described in the sections below.

The trend in the new model that is presented in Figure 11 and Figure 12 illustrates the estimated and the calculated roughness overlapped on the same graphs. Based on the R-squared and average relative error values, the model was performing well. R-squared values of 0.4% and 0.8% indicate that the model can explain a significant portion of the variance in the target variable, which is a positive sign. Additionally, average relative errors of 0.11% and 0.13% suggest that, on average, the model’s predictions are relatively close to the actual values, which is also a good sign.

The end scope of the neural network model was to generate and export a DLL model. The dynamic link library (DLL) is a type of file that contains reusable code and resources that can be used in other programs. In our case, the DLL is needed to create a software application that is able to predict the final roughness of the 3D-printed part.

The code for the new application was written in Python; Figure 13 and Figure 14 present the Graphical User Interface (GUI) of the application. The application is called Roughness Prototype Calculator. The application can be used for manufacturing processes of both milling and printing. The whole Python program written for this neural modeling is presented in Anex1.

The new theoretical values of the roughness estimated by the new neural network model (NNM), are presented in Table 5. A comprehensive comparison was conducted between the roughness values obtained from the linear regression model (LRM), which is represented by the Te. *R_a_*; the neural network model (NNM), represented by the New. Ne. *R_a_*; and the actual *R_a_* measurements of the test samples, represented by the Ef. *R_a_*.

Visualization of the trend in the comparison between the results generated by the two models and the measured roughness is illustrated in Figure 15 and Figure 16.

The validation of the new NNM was done by using the same set of experimental data presented in Table 4, both for printing and milling. In addition to the estimated values obtained by LRM, Table 5 also presents the estimated values obtained by NNM; the predictions from both modeling methods have been compared to the values measured on the probes.

Analyzing all the aspects presented and also comparing the two models, we considered that, for this particular case, the most suitable solution is to use the neural network model for optimization and prediction of the surface roughness.

## 5. Experimental Research to Test the New Process Parameters, Optimized for HME

The HME Prototype illustrated in Figure 17 was designed and manufactured by the authors. This explains why proper calibration and optimization of the process parameters was essential, to obtain the most out of this new hybrid equipment.

The whole optimization methodology presented in this paper could be applied for other prototypes, in order to find their optimal process parameters.

The hybrid manufacturing equipment prototype was used for this research. The prototype combined the printing operation with milling, as shown in Figure 17. The HME Equipment illustrated in Figure 17 was designed and manufactured by the authors; details about the constructions and characteristics of this equipment have been published in the article [12]. The main characteristics of this new hybrid equipment are presented below. The main structure of the HME consisted of of a Mini CNC and an FDM 3D printing head. Precise and accurate operation is of utmost importance for this particular type of equipment, as this guarantees that the manufactured parts meet specific dimensional requirements and align with the original design consistently. The equipment must possess the capability to produce parts with tolerances as low as 0.1 mm, necessitating precise control and movement of the materials involved. The primary advantages lie in its speed and efficiency, enabling the production of parts quickly and efficiently without compromising on quality. This necessitates high-speed printing mechanisms capable of generating multiple material layers in a short timeframe, while effectively managing substantial volumes of diverse materials.

Material compatibility represents another vital requirement, as the equipment should be able to work with a wide range of materials, allowing for versatility and adaptability to meet the specific needs of various projects. Customization is also crucial, allowing for the production of parts and objects with unique shapes, sizes, and features. To achieve this, the equipment must be adaptable to different design requirements and be able to employ various processes and techniques. User-friendliness is an essential aspect of this equipment, particularly considering that it should be easy to use even for individuals with limited or no prior experience in manufacturing. Furthermore, it should require low maintenance and easily replaceable parts to minimize downtime and maximize efficiency.

Moreover, the equipment’s final design must be cost-effective, affordable, and capable of producing parts and objects at a high rate without compromising on quality or accuracy. This ensures that it can be utilized for both prototyping and manufacturing purposes, serving as a cost-effective solution for companies of all sizes. One of the main challenges faced during the design of the new prototype was the integration of the milling equipment and the FDM extruder while keeping the assembly simple and avoiding excessive complexity. The solution was to combine both tools, the extruder, and the cutter spindle on the same *x*-axis but with a 10 cm offset. The base of the equipment was positioned on the *y*-axis, while the *z*-axis was attached to the *x*-axis to control the layer height and cutting depth during the manufacturing process.

In addition to the *x*-, *y*-, and *z*-axis, two special axes, A and B, were added to guide the milling and printing tools. The *x*-axis had a total range of motion of 300 mm, with 250 mm being the active range. The *y*-axis had a total range of motion of 180 mm, with an active range of 150 mm. The *z*-axis had a total range of motion of 45 mm, with an active range of 30 mm. The overall dimensions of the equipment were 450 mm (width) × 280 mm (depth) × 360 mm (height). These dimensions were chosen to provide a stable and solid chassis, as milling processes involve greater forces compared to printing processes. The chassis used in this prototype was sourced from a Mini CNC piece of equipment called VEVOR CNC 3018 Pro manufactured by Vevor company from Shanghai China.

The next step involved designing a new set of supports to mount the manufacturing tools on the *x*-axis. SolidWorks 2022 software was utilized to obtain the CAD model for this purpose. The 3D printing operation was performed using a Prusa I3 3D printer. Once the supports were manufactured, they were used to hold the manufacturing tools, and the subassemblies were mounted on the guiding system. For this type of construction, it was necessary to have two motors on the *z*-axis to control the two manufacturing tools independently.

The FDM extruder subassembly used in this prototype was a low-cost one and consisted of several components, including a Nema 17 stepper motor, an extruder gear, a hot end, a nozzle, a filament drive, and a cooling system. The stepper motor drove the extruder gear, which pushed the filament through the nozzle. The extruder gear gripped the filament and pushed it through the hot end, where it was melted and then extruded through the nozzle. The hot end included a heating element, a temperature sensor, and a cooling fan. The size of the nozzle played a significant role in the final surface finish and the overall print quality.

The filament drive system ensured a consistent flow of filament and comprised the extruder gear, a guide tube (PTFE tube), and a filament drive mechanism. A cooling system was employed to cool the filament after it was extruded, preventing deformation, and ensuring the printed object retained its shape.

The milling subassembly equipment included a spindle motor, a spindle housing, a tool holder, bearings, a cooling system, a drive system, and control electronics. The spindle motor was the main component responsible for rotating the cutting tool and performing the milling operation. It was a high-speed motor capable of reaching tens of thousands of rotations per minute (RPM)s.

To manage both the 3D printing and CNC milling processes, a sophisticated control system and software were required. The control system coordinated the operation of both processes, including the movements of the printer’s motors and the operation of the extruder and milling head. The control system utilized an Arduino UNO R3 microcontroller board and a Shield V3 designed to control both CNC and 3D printing. The Shield included sockets for five A4988 stepper drivers and was powered separately. It also featured additional filtering capacitors for each driver to ensure precise equipment movement. Cura was used as the user interface software for generating G-Code in the printing process, providing settings for print material and quality parameters. Grbl Control 3.6 software was used for controlling the CNC milling head, executing the G-code generated in Solid CAM. This software generated tool paths compatible with the printer’s positioning system, enabling real-time execution.

The prototype also included an offline control module, allowing for equipment control through its own interface, as shown in Figure 7. The G-code interpreter used was Merlin, which translated the G-code commands generated by the slicing software into appropriate motor control signals. The G-code interpreter handled complex instructions involving both 3D printing and milling. Regular firmware updates were considered important based on experience.

In addition to the control software for 3D printing, a separate software program was required to control the CNC milling head. This software was responsible for generating tool paths for the milling head to cut the part. It needed to generate tool paths compatible with the printer’s positioning system and capable of real-time execution.

The prototype of the HME illustrated in Figure 17 was manufactured and tested by the authors. This prototype of the new equipment needs to be calibrated and the process parameters, both for printing and milling, need to be optimized, in order to improve the surface quality and to reduce the material consumption. Usually, for commercially available equipment, the manufacturers recommend the optimal process parameters for different materials.

When a new prototype is designed and manufactured, a specific process optimization is required to improve the printing and milling output. That is why the aim of the research presented in this article is to optimize the process parameters and to calibrate and obtain professional results from this prototype.

The methodology presented in this article could be used by other researchers when a new manufacturing prototype is designed and manufactured.

Current testing methods for hybrid manufacturing equipment involve the usage of multiple parts to evaluate the different aspects of the system. For example, one test may focus on the dimension and geometrical tolerancing, and another test on surface quality on different surfaces. While these testing new methods can provide valuable insights, they can also be time-consuming and expensive. However, using multiple parts can introduce variability and uncertainty into the testing process, making it harder to draw final conclusions about the performance of the equipment. To find a solution to these limitations, there is a need for a new part that can meet all the criteria for testing and validating hybrid manufacturing equipment. This part would need to be versatile enough to test a wide range of materials and parameters. It would also need to be robust enough to withstand the rigors of testing over an extended period. The need for a new part to test and validate hybrid manufacturing equipment comes from a need and desire to improve the accuracy, efficiency, and cost-effectiveness of testing in this field. By developing a new part that can meet all the requirements for testing and validation, researchers and users can speed up the process of the development of new hybrid manufacturing technologies and improve the performance of existing systems.

For this reason, in this chapter, the aim was to further validate the parameter optimization methods. A new part was designed and manufactured. The design of the part was realized in SolidWorks. The design of the part is illustrated in Figure 18. The part was realized from PLM material. The equipment used to manufacture the part was the SP hybrid manufacturing prototype. The overall dimensions were measured with a caliper, and the roughness was measured with the same roughness measurement equipment as before. We design a new test part, illustrated in Figure 18, to be able to evaluate the performance of SP HME, as presented in Figure 19. The test part for HME was compared with the one from the technical drawing illustrated in Figure 19. Each surface of the test part was numbered, and both dimensional accuracy and surface roughness were measured against the process parameters obtained by the neural network modeling.

Roughness measurements were taken across specific areas of the complex part which were numbered from 1 to 16 and A–F as can be seen in Figure 18 and Figure 19 to assess surface irregularities, with values recorded at various printing angles to comprehensively capture the nuanced effects of surface topography as well as overall dimensions.

The specific test part illustrated in Figure 18 and Figure 19 was designed to test the capabilities of our new HME prototype. The optimization results obtained by using the presented mathematical modeling were applied and used for manufacturing these test parts in order to validate these sets of optimal parameters (five sets generated by the software). In this manner, the theoretical contributions brought by this article were tested on a practical complex test part illustrated in Figure 18. And the values measured for the surface roughness were similar to the values measured on the 5 probes illustrated in Figure 3 and Figure 4.

The advantage resulting from using such a complex model to validate the hybrid equipment is the variety of measurements that need to be checked. The measurements that can be performed on this model are as follows:1–4—roughness after milling operation;5–8, 12—roughness after printing;9, 13—roughness at 60°;10, 15—roughness at 45°;11, 16—roughness at 30°;14—roughness at 50°;A–F—general dimensions

The results of the measurement are presented in Table 6.

In the first section of the table, the targeted values were set at 0.9 mm for entries 1 to 4, and 0.65 mm for entries 5 to 16. The measured values for these entries varied slightly around the targeted values, indicating a relatively close match between the targeted and actual measurements. However, there were some variations observed, especially for entries 9, 10, 11, and 14, where the measured values deviated significantly from the targeted values.

In the second section of the table, entries 17 and 18 had a targeted value of 50 mm, while entry 19 had a targeted value of 90 mm. For these entries, the measured values were very close to the targeted values, indicating a high level of accuracy in the measurements. Entry 20 had a targeted value of 10 mm, and the measured value matched precisely, indicating an accurate measurement. Entry 21 had a slight deviation, with a measured value of 10 mm instead of the targeted value of 11 mm. Entry 22 had both the targeted and measured values at 10 mm, indicating an accurate measurement.

The table demonstrates the comparison between targeted and measured values for a range of parameters or dimensions. While most measurements closely matched the targeted values, there were some instances of significant deviations, highlighting potential areas for further investigation or improvement in measurement accuracy.

It is important to recognize and address some limitations that have influenced our research process and the interpretation of our results. In this section, we outline the constraints and challenges that are pertinent to our study and their potential impact.

Throughout the course of our research, we encountered certain methodological limitations that were not acknowledged. One significant constraint was the complexity of the hybrid manufacturing process itself. The interplay between 3D printing and milling introduces intricate interactions that are challenging to model accurately. As a result, the process parameters we established may not fully encapsulate the dynamic nature of the hybrid process.

The acquisition of accurate and representative data was essential for meaningful research outcomes. However, our study was confined to a specific set of materials and geometries due to practical considerations. Consequently, the generalizability of our findings to a broader range of materials or complex geometries might be limited. Furthermore, variations in material properties, environmental conditions, or equipment performance could introduce uncertainties in our results.

While we rigorously conducted tests on multiple parts to validate our optimization parameters, the size of our sample might still be considered modest in certain contexts. The inherent variability between individual parts could impact the robustness of our conclusions. Additionally, our study focused predominantly on one specific hybrid manufacturing prototype, potentially limiting the applicability of our findings to other equipment configurations.

Certain limitations were inherent to the nature of our research topic. Achieving a seamless integration of 3D printing and milling processes within a single hybrid manufacturing system brought challenges. These challenges included aligning the axes accurately and managing interactions between the two processes. Despite our best efforts, some residual inaccuracies in the manufacturing process may persist.

Future research endeavors could further explore these limitations and extend the boundaries of knowledge in this burgeoning field, but also other limitations like optimization across materials and extending the optimization methodology to encompass a wide range of materials and complex geometry and to know how the hybrid manufacturing process parameters interact with different materials and intricate designs to achieve optimal outcomes. Another future research could be a dynamic process modeling topic, to develop dynamic models that can predict the behavior of the hybrid manufacturing process in real time. This could involve incorporating sensors and feedback loops to adjust process parameters on the fly and ensure consistent quality.

## 6. Discussion

In the present study, relevance to the hybrid manufacturing process of polymer materials was shown, with a specific focus on PLA materials.

The experimental research demonstrates the viability of utilizing polymer materials, especially PLA, in the hybrid manufacturing process. The combination of 3D printing and milling operations resulted in notable enhancements to surface roughness and the overall accuracy of the PLA components. This aligns with our initial expectations and validates the potential of hybrid manufacturing for achieving higher component quality.

In the broader context of polymer manufacturing, this research holds significant promise. By showcasing the compatibility of PLA with hybrid technology, we contribute to the expansion of sustainable manufacturing practices. The potential of improving both surface roughness and accuracy holds implications for industries requiring precision in their products, such as electronics and medical devices.

Our study’s objectives were inherently fulfilled through the comprehensive analysis of the hybrid manufacturing process. The case study involving a complex PLA part provided evidence for the capabilities of the new HME.

The distinction between the neural network model (NNM) and the linear regression model (LNM) in predicting final roughness ($R_{a}$) is a notable outcome. The superiority of NNM highlights the advantages of leveraging advanced predictive modeling techniques.

While the presented results are promising, it is important to recognize the limitations of this study. The reliance on a small number of experiments might limit the generalizability of findings. To mitigate this, future research should consider expanding the dataset and exploring variations in experimental conditions. Additionally, further investigations could explore the broader implications of hybrid manufacturing for different types of polymers and materials.

Our study successfully demonstrates the feasibility of employing PLA in the hybrid manufacturing process, opening avenues for enhanced surface quality and accuracy. The case study validates our new hybrid manufacturing equipment’s potential and paves the way for the replication of this methodology. The findings reinforce the significance of advanced modeling techniques and underscore the potential for sustainable and precision-driven manufacturing practices.

The final results at a higher level could be compared with the results obtained from other researchers like D Grguraš et al. [9] and Mohamad El Mehtedi et al. [5]. They used a similar concept of hybrid manufacturing equipment. They manufactured samples on the same equipment—they were using milling to improve the surface roughness. With their optimized printing and milling parameters, they obtained a surface roughness on the parts between 1.96, using a nozzle size of 0.4 mm, and 2.18, using a nozzle of 1.1 mm. The other group of researchers succeeded in obtaining roughness from a range between 2 and 13 µm. Even though the equipment and the testing conditions were similar, due to unknown variables like equipment performance, tool specification, testing conditions, etc., the results cannot be compared in a proper way.

## 7. Conclusions

The combination of 3D printing and milling operations in the hybrid process led to improved surface roughness and dimensional accuracy of the PLA parts;The case study involving a complex PLA part effectively demonstrated the capabilities of our newly developed hybrid manufacturing equipment. Importantly, the methodology employed here has the potential to be replicated when optimizing similar artisanal hybrid setups for enhanced polymer component production;The neural network model (NNM) presented superior accuracy in predicting final roughness $\left(R_{a}\right)$ compared to the linear regression model (LNM);Through the utilization of optimized process parameters derived from these models, we achieved reliable estimations of the obtained roughness ($R_{a}$);The observed deviation of roughness ($R_{a}$) values from predicted estimates was both acceptable and consistently maintained across all samples tested;The innovative approach of integrating neural network modeling (NNM) with specialized Python programs for dynamic-link library (DLL) management provided favorable outcomes, effectively optimizing the functionality of our new hybrid manufacturing equipment (HME);A key highlight of this research is its ability to generate precise estimations for output parameters, particularly roughness ($R_{a}$), even when working with a limited number of experimental iterations;This study’s significance extends to other users of similar artisanal hybrid equipment, as they can apply our research methodology to enhance their equipment’s functionality. This involves identifying optimal process parameters to achieve improved equipment performance;In summary, the main goal of optimizing manufacturing parameters for enhanced surface roughness and dimensional accuracy of PLA components was effectively accomplished. The study’s findings not only validate the feasibility of employing PLA in hybrid manufacturing, but also highlight the potential for sustainable and precision-driven manufacturing practices in industries that demand high-quality components.

## Figures and Tables

**Figure 1 polymers-15-03610-f001:**
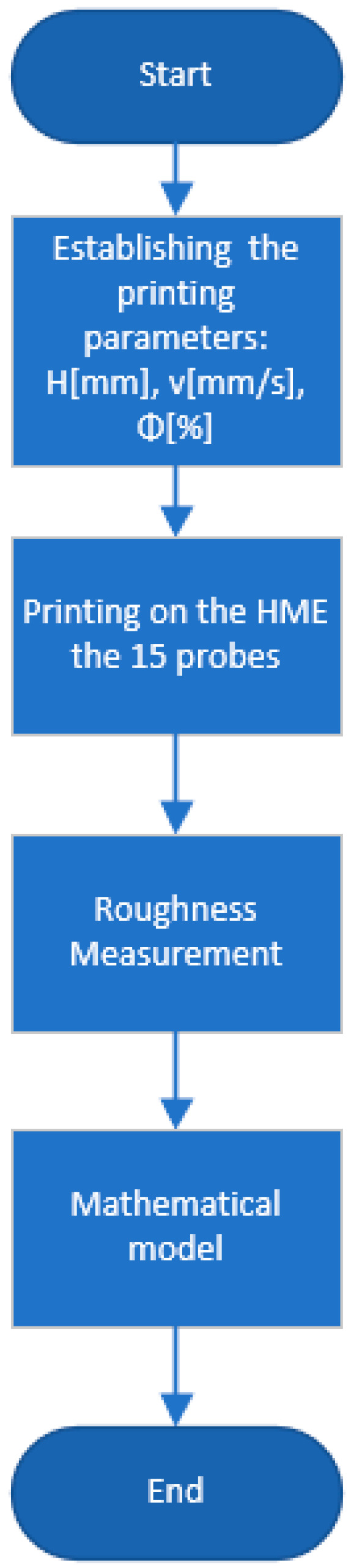
Workflow for printing operation.

**Figure 2 polymers-15-03610-f002:**
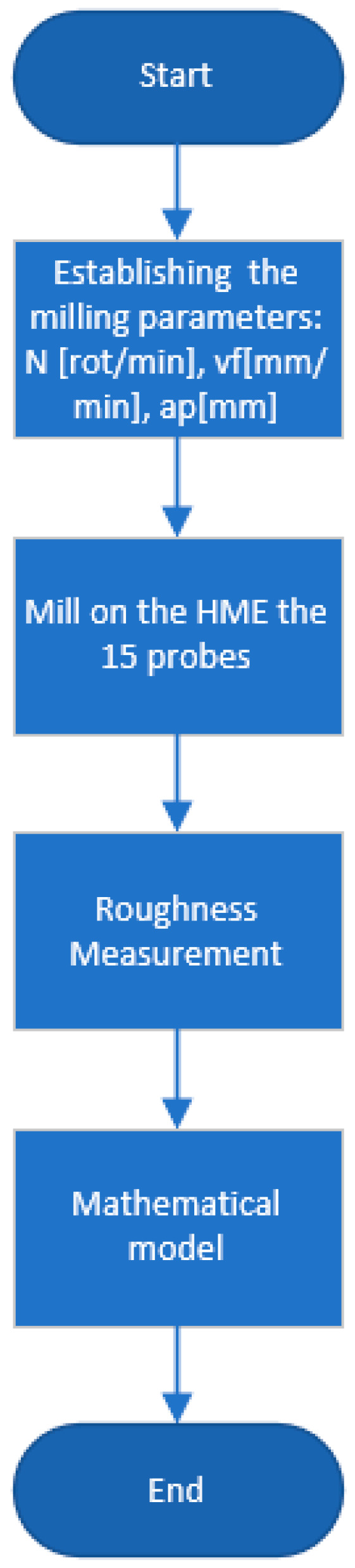
Workflow for milling operation.

**Figure 3 polymers-15-03610-f003:**
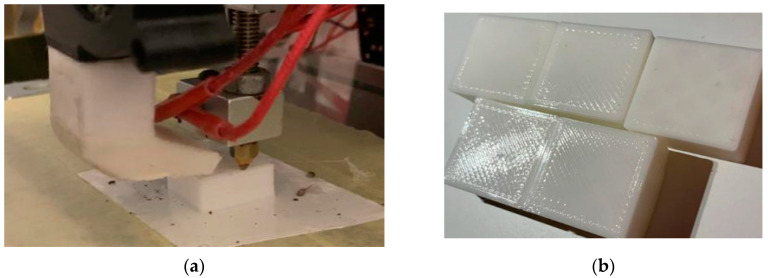
Printing operation on HME. (**a**) printing operation; (**b**) the printed samples from the HME operation.

**Figure 4 polymers-15-03610-f004:**
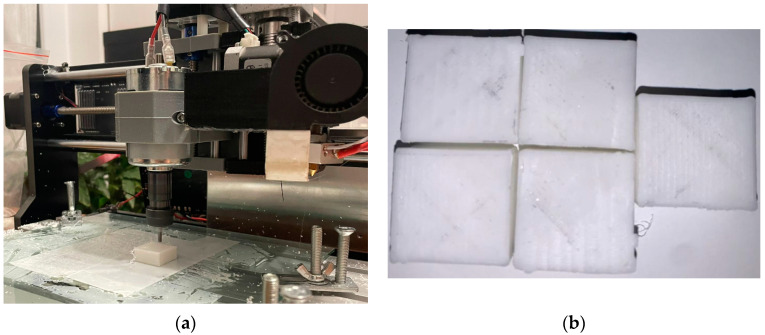
Milling operation on HME. (**a**) milling operation; (**b**) the milled samples from the HME operation.

**Figure 5 polymers-15-03610-f005:**
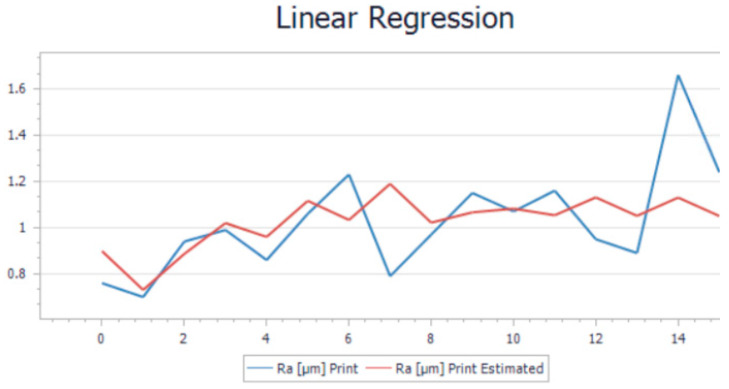
Linear regression trend for printing.

**Figure 6 polymers-15-03610-f006:**
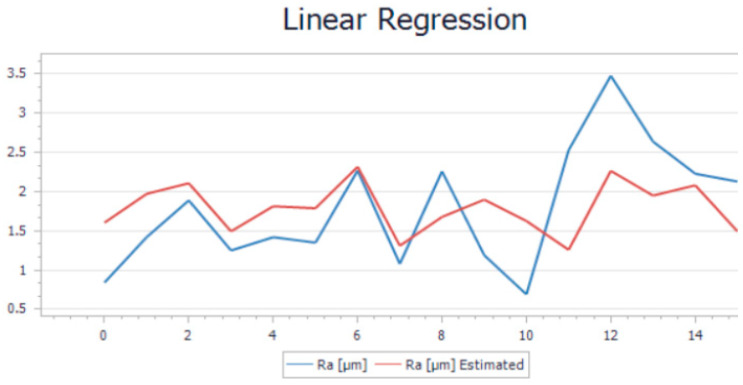
Linear regression trend for milling.

**Figure 7 polymers-15-03610-f007:**
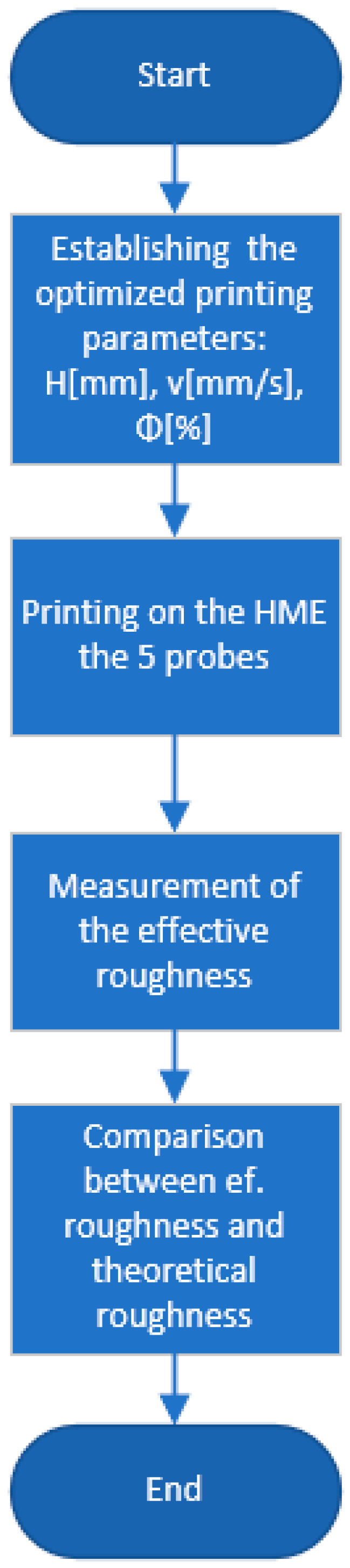
Workflow validation for printing operation.

**Figure 8 polymers-15-03610-f008:**
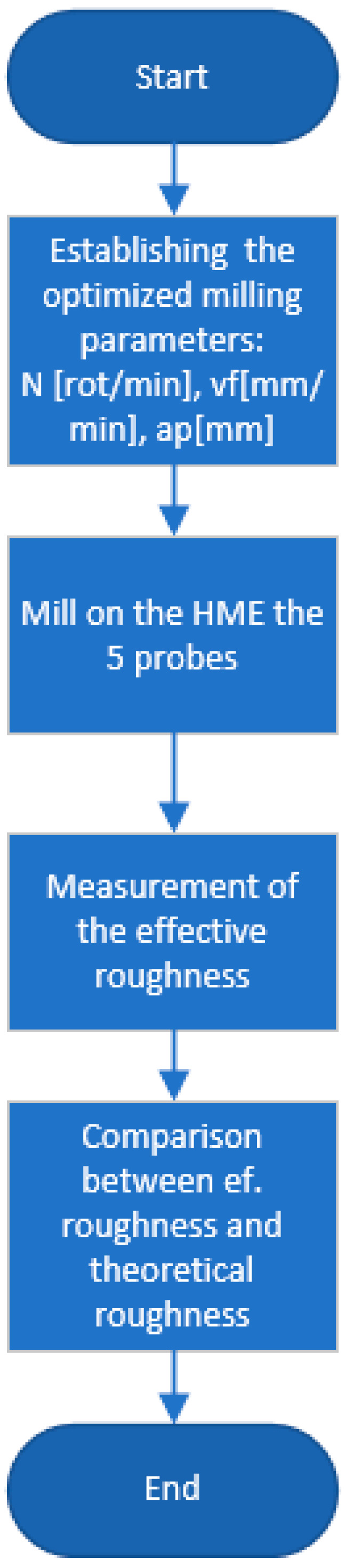
Workflow validation for milling operation.

**Figure 9 polymers-15-03610-f009:**
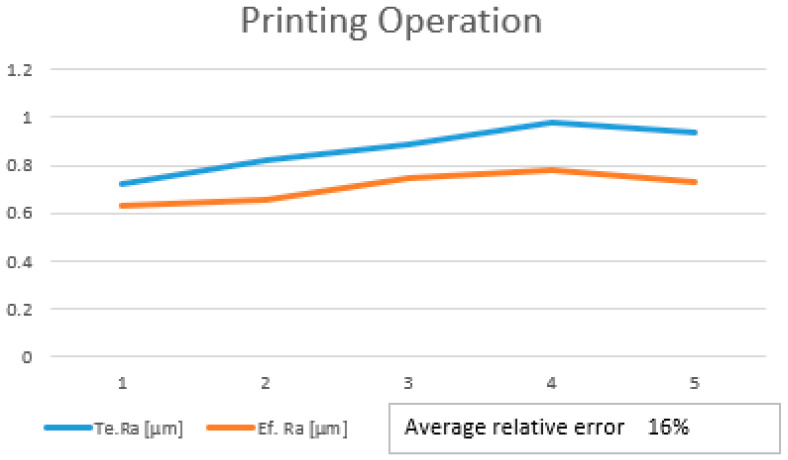
Validation trend for printing.

**Figure 10 polymers-15-03610-f010:**
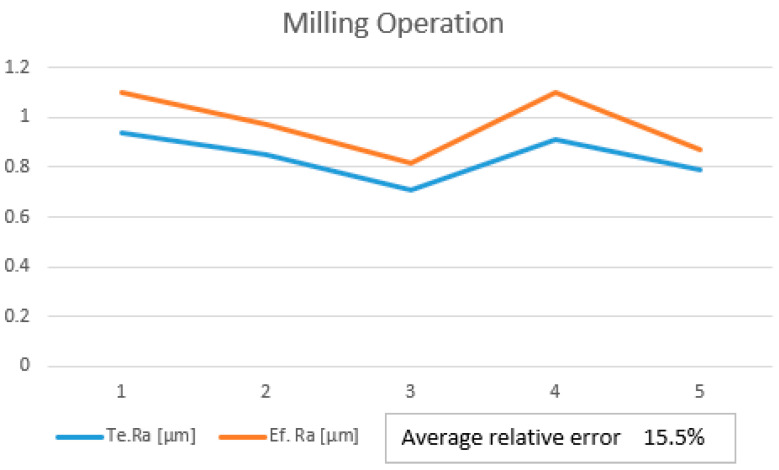
Validation trend for milling.

**Figure 11 polymers-15-03610-f011:**
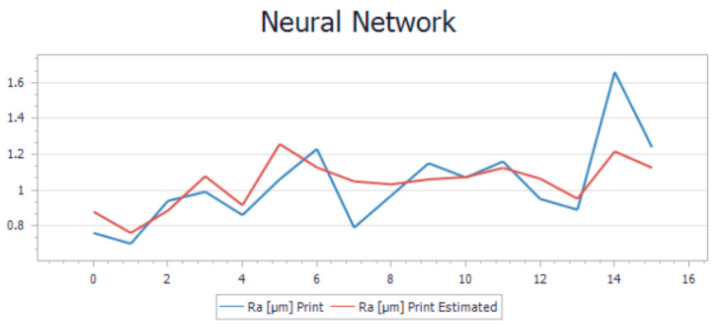
Neural network model trend for printing operation.

**Figure 12 polymers-15-03610-f012:**
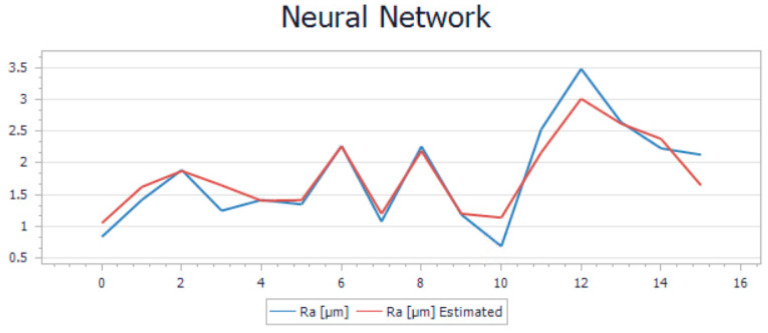
Neural network model trend for milling operation.

**Figure 13 polymers-15-03610-f013:**
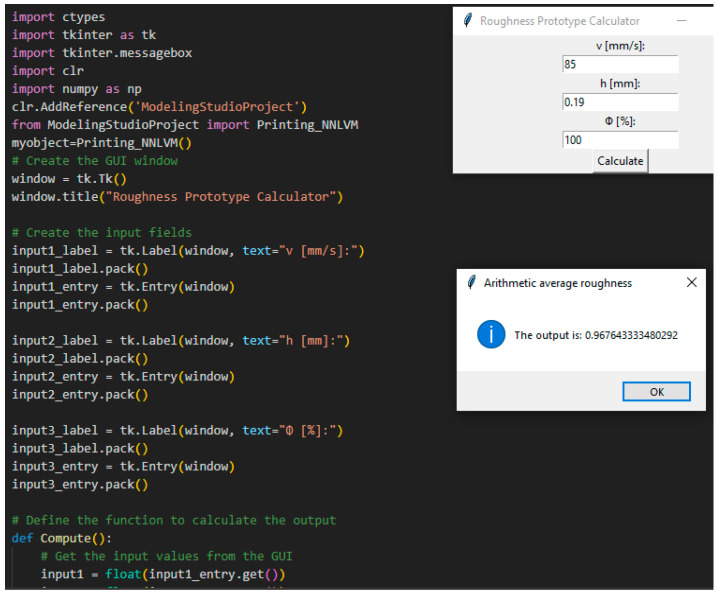
Roughness prototype for printing.

**Figure 14 polymers-15-03610-f014:**
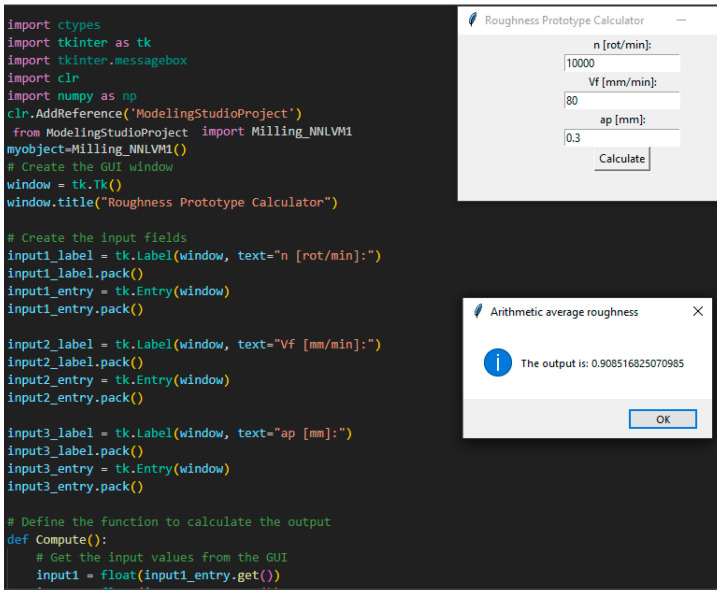
Roughness prototype for milling.

**Figure 15 polymers-15-03610-f015:**
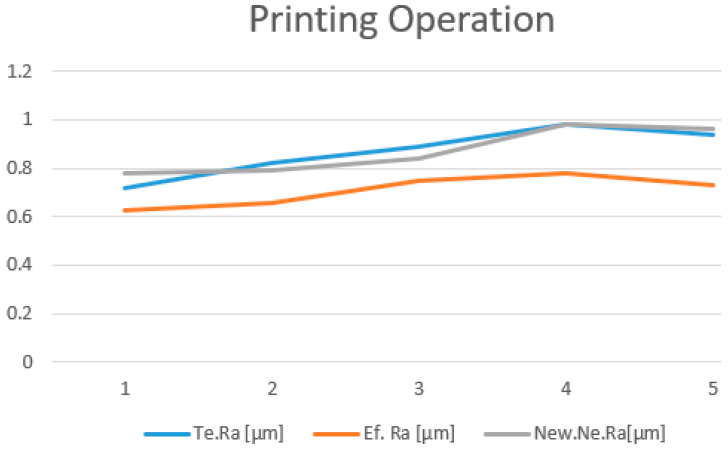
Roughness comparison for printing.

**Figure 16 polymers-15-03610-f016:**
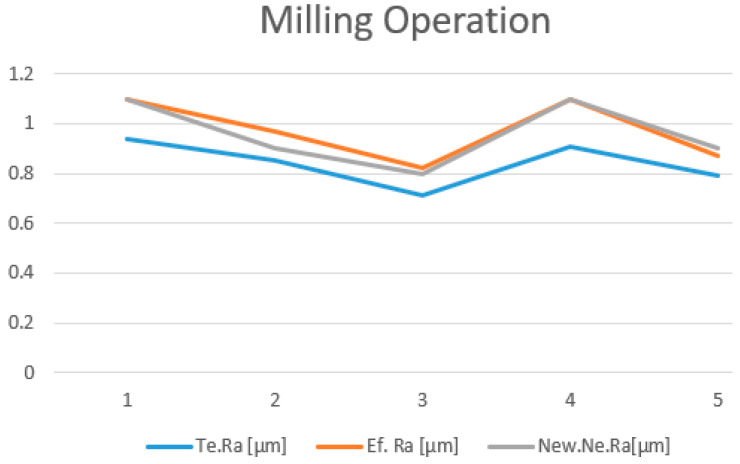
Roughness comparison for milling.

**Figure 17 polymers-15-03610-f017:**
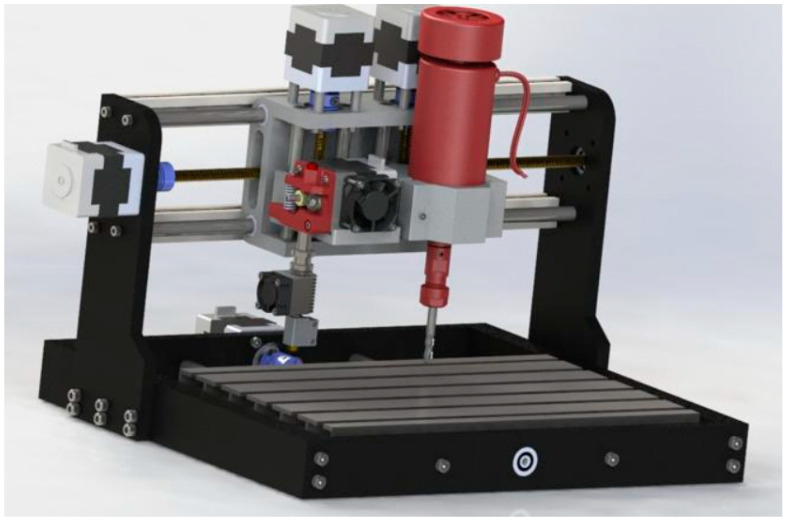
HME Prototype.

**Figure 18 polymers-15-03610-f018:**
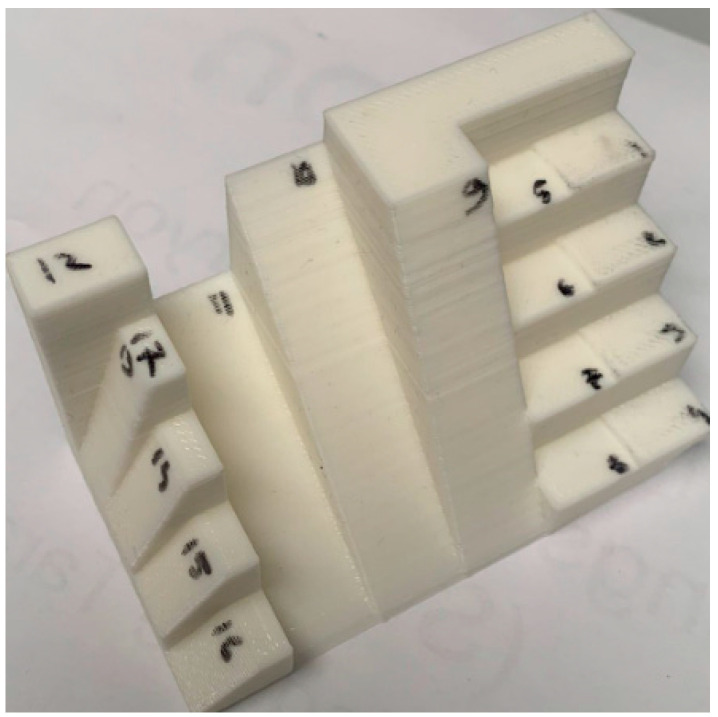
Newly designed validation part.

**Figure 19 polymers-15-03610-f019:**
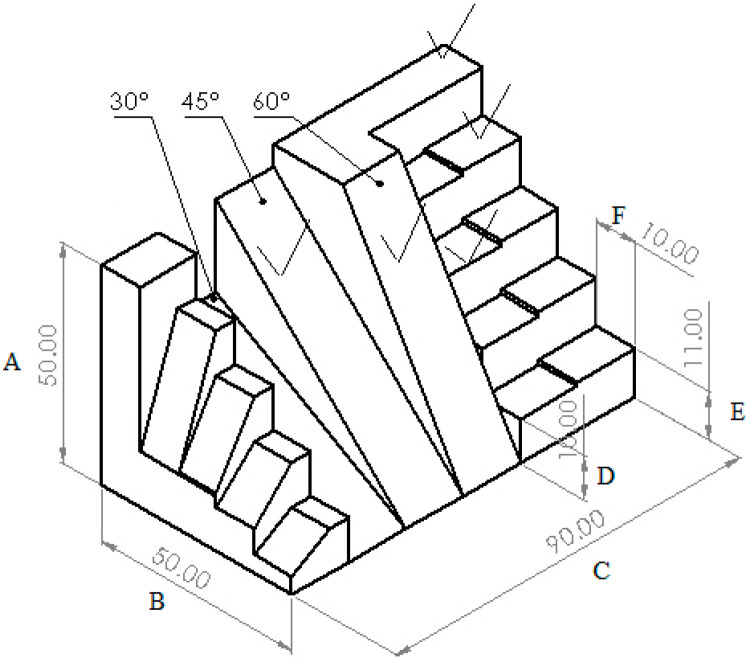
Newly designed validation part—3D drawing.

**Table 1 polymers-15-03610-t001:** List of technological parameters on 3 levels.

Level	n [rot/min]	vf [mm/min]	ap [mm]	v [mm/s]	h [mm]	Φ [%]
−1	500	80	0.1	30	0.2	80
0	5250	240	0.2	55	0.26	90
1	10,000	400	0.3	80	0.32	100

**Table 2 polymers-15-03610-t002:** Preliminary experimental results for printing.

Level	v [mm/s]	h [mm]	Φ [%]	$R_{a}$ [µm]
1	40	0.2	80	0.76
2	40	0.26	100	0.75
3	40	0.32	90	0.94
4	60	0.2	80	0.99
5	60	0.26	90	0.86
6	60	0.32	80	1.06
7	80	0.2	90	1.23
8	80	0.26	80	0.79
9	80	0.32	100	0.97
10	90	0.3	100	1.15
11	90	0.32	100	1.07
12	80	0.36	100	1.16
13	75	0.36	90	0.95
14	75	0.26	90	0.89
15	65	0.3	80	1.66
16	65	0.2	80	1.24

**Table 3 polymers-15-03610-t003:** Preliminary experimental results for milling.

Level	n [rot/min]	vf [mm/min]	ap [mm]	$R_{a}$ [µm]
1	500	240	0.2	0.84
2	10,000	240	0.2	1.42
3	5250	80	0.2	1.89
4	5250	400	0.1	1.25
5	5250	240	0.1	1.42
6	5250	240	0.2	1.35
7	10,000	80	0.1	2.27
8	500	400	0.1	1.08
9	10,000	400	0.1	2.26
10	500	80	0.3	1.19
11	10,000	400	0.3	0.69
12	500	400	0.3	2.53
13	10,000	80	0.3	3.48
14	500	80	0.1	2.64
15	5250	80	0.3	2.23
16	5250	400	0.1	2.13

**Table 4 polymers-15-03610-t004:** Optimized manufacturing parameters.

Optimized Parameter for Printing			Roughness Values		Optimized Parameter for Milling			Roughness Values	
v [mm/s]	h [mm]	Φ [%]	$\mathrm{Te}.\;R_{a}$ [µm]	$\mathrm{Ef}.\;R_{a}$ [µm]	n [rot/min]	vf [mm/min]	ap [mm]	$\mathrm{Te}.\;R_{a}$ [µm]	$\mathrm{Ef}.\;R_{a}$ [µm]
45	0.21	100	0.72	0.628	10,000	80	0.3	0.94	1.1
55	0.26	100	0.82	0.659	8000	100	0.2	0.85	0.97
65	0.28	100	0.89	0.748	6000	120	0.3	0.71	0.82
75	0.31	100	0.98	0.781	9000	90	0.2	0.91	1.1
85	0.19	100	0.94	0.730	7000	110	0.2	0.79	0.87

**Table 5 polymers-15-03610-t005:** *R_a_* comparison between the estimated values by LRM, the estimated values by NNM, and the measured values.

Results Comparison for Printing			Results Comparison for Milling		
Te. *R_a_* [µm]	Ef. *R_a_* [µm]	New. Ne. *R_a_* [µm]	Te. *R_a_* [µm]	Ef. *R_a_* [µm]	New. Ne.*R_a_* [µm]
0.72	0.628	0.78	0.94	1.1	1.1
0.82	0.659	0.79	0.85	0.97	0.9
0.89	0.748	0.84	0.71	0.82	0.8
0.98	0.781	0.98	0.91	1.1	1.1
0.94	0.73	0.96	0.79	0.87	0.9

**Table 6 polymers-15-03610-t006:** The outcomes of the part measurements conducted for equipment calibration purposes.

Nr.crt.	Targeted Value [µm]	Measured Value [µm]
1	0.9	0.83
2	0.9	0.89
3	0.9	0.81
4	0.9	0.81
5	0.65	0.62
6	0.65	0.65
7	0.65	0.74
8	0.65	0.73
9	0.65	1.13
10	0.65	1.49
11	0.65	2.43
12	0.65	0.66
13	0.65	1.5
14	0.65	1.26
15	0.65	1.30
16	0.65	1.19
17	50	49.5
18	50	49.8
19	90	90
20	10	10
21	11	10
22	10	10

## Data Availability

Not applicable.

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
