# Peer review of "Process Parameter Optimization for Hybrid Manufacturing of PLA Components with Improved Surface Quality"

_polymers, 2023, doi:10.3390/polym15173610_

Round 1

Reviewer 1 Report

The author has proposed a new, improved method to develop 3D-printed parts with improved surfaces in the manuscript. It is a very interesting work, but it is required to incorporate suggestions/comments to improve the manuscript further.

1. Add a few lines of obtained results in the abstract.

2. The author should define all abbreviations the first time, such as HM and many others not defined the first time. look carefully.

3. The author should elaborate on the objective of the work for more clarity.

4. In Fig 1 and 2, Fig 7 and 8, enlarge the font, nothing is visible.

5. In Fig 4 and 5, mention a and b for better clarity

6. Enlarge the scale of Fig 6 and 7. 

7. The author should mention the data set in the Neural network (80 &20 or 60 &40, etc). It is missing.

8. Fig 13 and 14 should be more clear. it is not visible.

9. The author has only presented the results. The author should add the discussion part.

10. Conclusion should be in point form.

Author Response

Please see the attached file, presenting the responses to reviewer 1. Thank you!

Responses to reviewer 1:

  1. Add a few lines of obtained results in the abstract.
    • Thank you for the advice. We added the lines 19-25 in the abstract.
  2. The author should define all abbreviations the first time, such as HM and many others not defined the first time. look carefully.
    • We defined now the abbreviations before first time when they were used (e.g. lines 59, 85, 90, etc.). Thank you!
  3. The author should elaborate on the objective of the work for more clarity
    • We elaborated the specific objectives of the work, which are presented now in lines 66 – 78. Thank you!
  4. In Fig 1 and 2, Fig 7 and 8, enlarge the font, nothing is visible.
    • The font size was increased, to be readable. Thank you.
  5. In Fig 4 and 5, mention a and b for better clarity
    • They have been distinguished, thank you.
  6. Enlarge the scale of Fig 6 and 7. 
    • The scaled was enlarged, thank you.
  7. The author should mention the data set in the Neural network (80 &20 or 60 &40, etc). It is missing.
    • We mentioned now the data set in the Neural network, in lines 354-361.
  8. Fig 13 and 14 should be clearer. it is not visible.
    • We enlarged these figures, thank you.
  9. The author has only presented the results. The author should add the discussion part.
    • We added a discussion part (lines 640-670), thank you.
  10. Conclusion should be in point form.

Thank you very much for the professional evaluation and useful suggestions you gave us, on how to improve the article.

Reviewer 2 Report

Dear Authors, the manuscript ‘Process parameters optimization for hybrid manufacturing of PLA components, with improved surface quality’, Manuscript ID: polymers-2563521, have some weakness, listed further, that must be revised in a required manner.

Please find below the most crucial comments:

1.      Reading the ‘Abstract’ section it is generally poor. The Authors did not provide the main advantage of the results proposed. This should be emphasized.

2.      The main motivation of the work is literally hidden, the sentences ‘Despite these limitations, milling and FDM printing are still valuable manufacturing techniques, as they offer unique advantages that other techniques do not provide. It is important to choose the right technique based on the specific requirements of the application.’, lines 50-52, do not derives from the lack in the current state of knowledge.

3.      The critical review, especially for the literature review, is not provided. The Authors should indicate what is missing in the studies already presented. Introducing new materials or technologies must be argued by some lacks in knowledge.

4.      The structure of the first two sections should be modified. Taking all of the information presented into consideration, the Authors should first introduce the main line of the study then proposed some state-of-the-art data and, finally, add some information on the purpose. Now it is some ideas presented and then a literature review which is insufficient in that way. Replace and modify those sections.

5.      Still, for the first two sections, the number of literature review items is low. If the Authors propose any ‘State-of-the-art’ reviewing, this must be provided with more deep studies and research. Looks like the Authors concentrate only on the required area, which gave them a suitable response but did not comprehensively study the lack in the field.

6.      Considering section no.3, the flow charts of the process are unclear. The quality of Figures 1 and 2 is too poor that not allow the Reader to understand the process.

7.      The values of the parameters of the process should be mentioned, not only reflecting the previous studies: ‘The parameters that were chosen to be studied and collect data from them were chosen based on preliminary experiments performed on separate printing and milling equipment, experience, and other studies [11].’, lines 174-176. Any justification should be added as well, like in Table 1.

8.      Considering the roughness measurement, mentioned in lines 211-213: ‘The roughness measurements were performed with a Mitutoyo Contracer CP - 200 / 400 - Contour Measuring Instrument.’, there are no details on it. What about the measurement parameters? The Authors should add some details on the measurement parameter values that can significantly affect the results obtained.

9.      Similar to the previous comment, there are no words against measurement uncertainty, noise or other errors. Please referee to some valuable information more comprehensively, as in the examples:

(1)   https://www.doi.org/10.1088/2051-672X/3/3/035004

(2)   https://www.doi.org/10.3390/ma16051865

(3)   https://www.doi.org/10.1117/1.OE.59.6.064110

10.  The disacussion in section 5 must be more emphasized. In practice, there are no limitations presented for the purpose. Usually, the Authors propose some prospects to resolve some additional or, respectively, not fully resolved issues. In the reviewed manuscript Authors did not propose any further studies.

11.  The ‘Conclusion’ section should be divided into separated and numbered gaps. Firstly. The main advantage included in the purpose must be strongly separated from detailed information. Secondly,

12.  Additionally, the quality of many Figure is extremely poor, like Figures 5, 6, 7, 8, 10, 11, 12, 13, 14, 15 and 16. In fact, all of the Figures require strong improvements.

Generally, the proposed area of study can be classified as interesting and up-to-date, respectively, the manuscript, at least in the current form, has some strong weaknesses and is not suitable for publication in a quality journal as the Polymers is, it must be improved significantly before any further processing, if allowed by the Editor.

Author Response

Please see the attached file, presenting the responses to reviewer 2. Thank you!

Responses to reviewer 2:

  1. Reading the ‘Abstract’ section it is generally poor. The Authors did not provide the main advantage of the results proposed. This should be emphasized.
    • The main advantages of the results proposed are emphasized within the lines 19 - 31 of the abstract. Thank you for the suggestion.
  2. The main motivation of the work is literally hidden, the sentences ‘Despite these limitations, milling and FDM printing are still valuable manufacturing techniques, as they offer unique advantages that other techniques do not provide. It is important to choose the right technique based on the specific requirements of the application.’, lines 50-52, do not derives from the lack in the current state of knowledge.
    • The motivation of the work is presented in new added lines 63 – 78, where the specific objectives of this research are presented as well. Thank you for suggesting this clarification.
  3. The critical review, especially for the literature review, is not provided. The Authors should indicate what is missing in the studies already presented. Introducing new materials or technologies must be argued by some lacks in knowledge.
    • This article is introducing a newly developed HME designed by the authors, for producing complex PLA parts. The prototype of this new HME has a specific rigidity and stability, different as compared to separate milling machines or individual 3D printers.
    • There are few HME commercially available, such as 5-axis works (https://5axismaker.co.uk/ ), which are able to do both printing and milling of a PLA component, but we did not find relevant publications presenting how the process parameters have been optimized for these new HME or how the surface quality of the printed parts could be improved when using such a new HME.
    • Few companies, who already developed their new HME equipment, keep confidentiality on the calibration procedures and specific methods of optimizing the process parameters of their HME. That is why, this kind of information is missing within the studies already published, such as:
      1. Grguraš, Damir & Kramar, D.. (2017). Optimization of Hybrid Manufacturing for Surface Quality, Material Consumption and Productivity Improvement. Strojniški vestnik - Journal of Mechanical Engineering. 63. 567-576. 10.5545/sv-jme.2017.4396.
      2. Jayant Giri, Pranay Shahane, Shrikant Jachak, Rajkumar Chadge, Pallavi Giri, Optimization of FDM process parameters for dual extruder 3d printer using Artificial Neural network, Materials Today: Proceedings, Volume 43, Part 5, 2021, Pages 3242-3249, ISSN 2214-7853, https://doi.org/10.1016/j.matpr.2021.01.899.
  • Dinesh Yadav, Deepak Chhabra, Ramesh Kumar Garg, Akash Ahlawat, Ashish Phogat, Optimization of FDM 3D printing process parameters for multi-material using artificial neural network, Materials Today: Proceedings, Volume 21, Part 3, 2020, Pages 1583-1591, ISSN 2214-7853, https://doi.org/10.1016/j.matpr.2019.11.225.
  1. Tadeusz Mikolajczyk, Tomasz Malinowski, Liviu Moldovan, Hu Fuwen, Tomasz Paczkowski, Ileana CiobanuCAD CAM System for Manufacturing Innovative Hybrid Design Using 3D Printing, Procedia Manufacturing, Volume 32, 2019, Pages 22-28, ISSN 2351-9789, https://doi.org/10.1016/j.promfg.2019.02.178

  1. The structure of the first two sections should be modified. Taking all of the information presented into consideration, the Authors should first introduce the main line of the study then proposed some state-of-the-art data and, finally, add some information on the purpose. Now it is some ideas presented and then a literature review which is insufficient in that way. Replace and modify those sections.
    • We have improved the structure of the first two sections. Motivation is presented (pages 61-63) and the specific objectives of this research have been underlined (pages 64-78), related to the state of the art analysis.
  2. Still, for the first two sections, the number of literature review items is low. If the Authors propose any ‘State-of-the-art’ reviewing, this must be provided with more deep studies and research. Looks like the Authors concentrate only on the required area, which gave them a suitable response but did not comprehensively study the lack in the field.
    • Thank you for your professional evaluation and for your time dedicated to evaluate our research. You are right, we did not present a comprehensively study of the lack in the field. This is a research article, not a reviewing article, that is why we analyzed and made reference only to the specific articles focused on similar studies and targeting similar topics.
    • There are plenty of articles presenting machining by milling of the PLA components, or plenty of publications presenting the FDM process using PLA and ABS filaments, but all these publications are not relevant for our research because their findings are referring to separate operations, using totally different equipment.
    • The specific results we obtained have been compared (in lines 671-680) to:
      1. Grguraš, Damir & Kramar, D.. (2017). Optimization of Hybrid Manufacturing for Surface Quality, Material Consumption and Productivity Improvement. Strojniški vestnik - Journal of Mechanical Engineering. 63. 567-576. 10.5545/sv-jme.2017.4396.
      2. Mehtedi, M. E., Buonadonna, P., Carta, M., Mohtadi, R. E., Marongiu, G., Loi, G., & Aymerich, F. (2023). Effects of milling parameters on roughness and burr formation in 3D- printed PLA components. Procedia Computer Science, 217, 1560–1569. https://doi.org/10.1016/j.procs.2022.12.356
    • Anyhow, even these previous researches found closer to our topic, were done using different hybrid equipment, with different testing conditions and a slightly different strategy of research. The results obtained by us are good, as compare to the results presented in these publications.

  1. Considering section no.3, the flow charts of the process are unclear. The quality of Figures 1 and 2 is too poor that not allow the Reader to understand the process.
    • The flow charts were enlarged, thank you.
  2. The values of the parameters of the process should be mentioned, not only reflecting the previous studies: ‘The parameters that were chosen to be studied and collect data from them were chosen based on preliminary experiments performed on separate printing and milling equipment, experience, and other studies [11].’, lines 174-176. Any justification should be added as well, like in Table 1.
    • Thank you for the suggestion We added lines 204-212
  3. Considering the roughness measurement, mentioned in lines 211-213: ‘The roughness measurements were performed with a Mitutoyo Contracer CP - 200 / 400 - Contour Measuring Instrument.’, there are no details on it. What about the measurement parameters? The Authors should add some details on the measurement parameter values that can significantly affect the results obtained.
    • Measurement parameters have been added. (lines 249-252). Thank you.
  4. Similar to the previous comment, there are no words against measurement uncertainty, noise or other errors. Please referee to some valuable information more comprehensively, as in the examples:
    1. (1) https://www.doi.org/10.1088/2051-672X/3/3/035004
    2. (2) https://www.doi.org/10.3390/ma16051865
  • (3) https://www.doi.org/10.1117/1.OE.59.6.064110
  • That is right, we did not present possible errors caused by noise or uncertainties. These types of errors have a significant influence mainly when measuring small values of the roughness, such as for grinded surfaces, where the surface roughness is about 1-2 µm. For PLA parts made on the HME, the surface roughness is considered acceptable in the range of 10-20 µm.
  1. The discussion in section 5 must be more emphasized. In practice, there are no limitations presented for the purpose. Usually, the Authors propose some prospects to resolve some additional or, respectively, not fully resolved issues. In the reviewed manuscript Authors did not propose any further studies.
    • These aspects have been emphasized now (lines 603-637). Thank you.
  2. The ‘Conclusion’ section should be divided into separated and numbered gaps. Firstly. The main advantage included in the purpose must be strongly separated from detailed information. Secondly,
    • The Conclusions section has been improved, by pointing out the gaps filled by our findings. Bullets have been used, as suggested by reviewer 1.
      1. The main goal of this research has been achieved and the manufacturing parameters were optimized enhanced surface roughness quality. Also, better dimensional accuracy was effectively accomplished, for the PLA components made on our new HME.
    • The lines 708-712 points out the overall conclusion
    • Other conclusions with more information about the other findings are presented in lines 685-707
  3. Additionally, the quality of many Figure is extremely poor, like Figures 5, 6, 7, 8, 10, 11, 12, 13, 14, 15 and 16. In fact, all of the Figures require strong improvements.
    • All the images have been enlarged. Thank you!

Generally, the proposed area of study can be classified as interesting and up-to-date, respectively, the manuscript, at least in the current form, has some strong weaknesses and is not suitable for publication in a quality journal as the Polymers is, it must be improved significantly before any further processing, if allowed by the Editor.

Thank you very much for the professional evaluation, which pointed out the missing and week points of our article, which was now significantly improved and hopefully suitable for publication.

Round 2

Reviewer 1 Report

No Comments

Reviewer 2 Report

The manuscript was improved suitably and can be further processed by the Polymers journal.